# Evaluating Rural Health Disparities in Colombia: Identifying Barriers and Strategies to Advancing Refugee Health

**DOI:** 10.3390/ijerph20206948

**Published:** 2023-10-20

**Authors:** John Diaz, Isabel Taboada, Adriana Abreu, Lara Vargas, Ysabel Polanco, Alex Zorrilla, Norman Beatty

**Affiliations:** 1Gulf Coast Research and Education Center, University of Florida, Plant City, FL 33563, USA; 2College of Medicine, University of Florida, Gainesville, FL 32610, USA; isabeltaboada@ufl.edu (I.T.); adrianaabreu@ufl.edu (A.A.); lvargaslavarez@ufl.edu (L.V.); alex.zorrilla@ufl.edu (A.Z.); norman.beatty@medicine.ufl.edu (N.B.); 3Faculty of Medicine, University of Antioquia, Medellin 050010, Antioquia, Colombia; ysabel.polanco@udea.edu.co; 4Emerging Pathogens Institute, University of Florida, Gainesville, FL 32610, USA

**Keywords:** Delphi technique, health equity, displaced communities, consensus

## Abstract

Health disparities within rural communities, notably those affecting migrant and refugee populations, are well-documented. Refugees often grapple with high disease burdens and mortality rates due to limited access to primary healthcare and their vulnerable socio-economic and political situations. This issue is particularly acute in the rural areas around Medellin, Colombia, where the refugee influx exacerbates the existing public health challenges. Studies highlight a substantial gap between community needs and public health policies, resulting in inadequate healthcare access. Our study, utilizing the Delphi technique, aimed to identify common barriers and strategies to enhance rural healthcare for refugees. Through consensus-building with community leaders, we identified six primary barriers to healthcare access and five barriers to healthcare quality. Community leaders endorsed five strategies to address the access barriers and eight strategies to improve healthcare quality. This research provides valuable insights for optimizing resource allocation and designing effective support programs for these vulnerable populations.

## 1. Introduction

We live in a world that is exponentially becoming interconnected, and, with many governments suffering from political instability, countries are looking at drastic demographic changes due to the establishment of new global migration patterns, with 280.6 million displaced immigrants worldwide [1]. This has forced many countries to quickly adapt and create new migration laws, and healthcare access is at the forefront of the principal issues that need to be tackled. One of the countries currently facing many of these issues is Colombia, since, as of July 2018, there were over one million Venezuelans residing in Colombia and about half of them through irregular migration [2]. The International Organization for Migration (IOM) characterizes irregular migration as “the movement occurring outside the established regulatory norms of the country of origin, the transit countries, and the destination country” [3]. Immigration from neighboring countries such as Venezuela exacerbates existing disparities that come from migration patterns that occur within the country, with the main case being that of people who are displaced due to armed conflict. Reports from February 2020 indicated that the national agency responsible for assisting those affected by conflict, Unidad para la Atención y la Reparación a las Víctimas (UARIV), had recorded approximately 8 million people internally displaced by violence or the threat of violence, which is equivalent to 16% of Colombia’s population [4]. While the displaced populations in this country face issues related to the possibility of natural disasters occurring in the areas they are forced to live in, as well as the lack of established infrastructure and basic necessities available in their new homes—which occurs because a lot of these houses were unofficially settled [4]. The immigrant community in Colombia faces healthcare barriers mainly due to their lack of legal status and, thus, of insurance acquisition [5]. Current research has focused on these migration patterns and how different populations are affected by these circumstances and has identified various barriers and challenges. There is still a lack of general agreement on what the primary barriers are that directly affect these populations. The purpose of our research is to identify a consensus on these barriers and establish potential strategies for advancing healthcare access and the quality of care for the refugee populations of Medellin, Colombia.

### 1.1. The Refugee Crisis in Colombia

The refugee crisis in Colombia has escalated in recent years due to the economic and political collapse in its neighboring country, Venezuela. This economic and political instability in Venezuela has led to many families deciding to move to other areas of Latin America to temporarily seek refuge until their nation is able to recover—over 7 million Venezuelans have fled the country, with more than 2.4 million seeking to settle in Colombia [6]. This mass migration of refugees seeking asylum has led to a public health crisis in the rural areas surrounding major cities in Latin America, as accessibility to legal housing options is limited. Informal settlements were already experiencing systematic constraints due to a lack of infrastructure, and this is now further exacerbated by the continued influx of low-resource families and individuals who also need a place to live. Furthermore, access to resources and knowledge of government policy are widely unknown to these refugees, which also exacerbates the vulnerability of their situation, with access to healthcare being the leading issue [6].

Another important issue to note is that Venezuelan migrants are not the only refugee population in Colombia. Colombia has the second-highest number of internally displaced people in the world: a total of 5.6 million people. Nearly 90% of the country’s internally displaced population has been displaced from rural small towns to urban areas due to conflict and violence [7]. These internally displaced people move to urban areas in search of economic opportunities and a more stable life away from violence, leading to the creation of informal urban settlements which are a last resort for many families. Medellin has a tumultuous past, filled with violence and the marginalization of communities at the will of the drug cartels, and criminal activity is distinctly concentrated in the surrounding urban areas. The refugee populations in Medellin are constituted of immigrants and native Colombians who are internally displaced by these conflicts. As people are forced to leave their homes, they are faced with having to find other places to live, and end up in unstable housing situations, where hazards are the norm, putting their health at risk [8]. The typical health dangers that individuals face in the area stem from the physical environment, including landslides, floods, and urban fire hazards [4].

### 1.2. Immigrant Healthcare Access and Quality of Care

The Colombian government created many programs to combat the overwhelming number of refugees seeking asylum in Colombia. Not only are there physical barriers (e.g., a lack of good-quality transportation infrastructure and a lack of localized health centers) for refugees due to a multitude of factors, but their lack of legal residence in Colombia also makes it difficult for them to be seen by any medical professionals. There are two major programs in place to tackle this, which are the Permiso Especial de Permanencia (PEP) and the Estatuto Temporal de Protección para Migrantes Venezolanos (ETPMV). The Permiso Especial de Permanencia (PEP) was created in 2018 as a form of identification for Venezuelan migrants, allowing refugees to be considered under the label of “legal” rather than as intruders. This granted them temporary access to healthcare, education, work, and childcare with an emphasis on “temporary” as this was not created under the assumption that these people would intend to become citizens after the 2-year period was over. According to the rules and regulations of the PEP set up by the Colombian government, these privileges can be revoked if one is caught inappropriately using the PEP, if crimes are committed, and if they leave the country for more than 90 days [9]. The goal of the ETPMV is to allow migrants from Venezuela to have ten years to acquire a visa. By having more migrants enroll in this program, it ensures that the government is aware of their existence and can attempt to integrate them into citizenship while waiting for their residency to become properly authorized. According to the Colombian embassy, by helping refugees that are in irregular situations, which is statistically more than half, there is not only the potential to help the migrants themselves, but to help ensure that the country itself is not being set back by the complexities brought by such a large influx of migration [10]. This implies that these migrants will later be authorized to tap into other social safety nets set in place by the Colombian government, such as healthcare. Although the goal of these programs was to create better conditions for refugees, with more access to the social systems necessary to thrive, the information and spread of knowledge is lacking and many refugees have not obtained legal documents. According to a study conducted in 2022, less than half of all Venezuelan migrants have enrolled in PEP, and as of May 2021 only 383,000 Venezuelans were registered to begin the ETPMV process [5]. Furthermore, since most Venezuelan migrants have not obtained legal documents, many are uninsured and are forced to pay for health services out of pocket. Migrants who do not have health insurance are limited only to emergency healthcare services, only when deemed life-threatening or to special health services such as for pregnant women and children.

Since healthcare access is contingent on migratory status, it is evident that there are many gaps in access for refugees. According to a study which analyzed the connection between social determinants of health and the use of healthcare, those in particularly vulnerable populations, such as those who are of a low-income socioeconomic status, have less knowledge about current healthcare systems and rights [11]. With enrollment to these temporary migratory statuses, refugees are integrated into the natural-born citizen’s insurance system—either the subsidized or contributive system. The subsidized system is for those who are conventionally unemployed, meaning they do not have a secure sense of income and/or live below the poverty line. The contributive system is then taken on through formal employment. In addition, people in the contributive system can also pay for additional private health insurance for a faster medical service, especially with regard to medical specialists and medical procedures [12]. In addition to these bureaucratic barriers, it is important to note that there are significant physical barriers that impede access to healthcare. According to a quarterly report conducted in Colombia by the Danish Refugee Council, the three largest barriers to healthcare are a lack of documentation (46.71%), distance to health centers (12.50%), and access to transportation (7.40%) [13].

As access to healthcare for refugees is wavering and unstable, the quality of that healthcare becomes compromised. Research from multiple contexts in both the developed and developing world has established that migrants, particularly forced migrants, tend to exhibit greater disparities compared to host populations, both in terms of their state of health and access to good-quality services [14,15,16,17,18,19]. These disparities range from a lack of access to shelter to potable water, and the attainment of legal status. It is well known that migrants and refugees are more likely to experience both inconsistent access to and a lower quality of medical care, due to a multitude of different barriers [11,12,13]. As aforementioned, when refugees are enrolled in programs such as the PEP or ETPMV, they are allowed to enter either the contributive or subsidized insurance systems, depending on the program they qualify for. Those in the privatized healthcare insurance scheme have more resources allocated for their care, thus increasing their quality of care.

Inconsistent healthcare makes it difficult for refugees to have their health problems attended to in a timely manner. According to a study on the healthcare obstacles faced by Venezuelan migrants in Colombia in 2021, the lack of local entities’ knowledge about the needs of the community aggravates the health disparities and directly leads to inconsistent care while decreasing the overall quality of care [20]. The Colombian government has made efforts to create more access to quality healthcare through their universal healthcare programs, but this has presented barriers of its own. For example, it is stated in their legal system that foreigners who enter Colombia without health insurance, no matter the situation, are entitled to emergency care. The lack of an explicit definition of what constitutes an emergency allows xenophobia and discrimination to affect refugees’ access to health. There are instances of Venezuelan migrants being completely turned away from services due to xenophobia, playing a direct role in not only quality but overall access of migrants to healthcare as well [10,21]. In addition to other general barriers that refugee populations face, being denied access to quality healthcare exacerbates these barriers.

### 1.3. The Role of Extension Organizations and Achieving Community Consensus

Extension organizations play a pivotal role in empowering communities by bridging the gap between formal knowledge and practical application, equipping individuals with relevant skills and information to improve their livelihoods [22]. Extension education represents a non-formal education mechanism that university institutions, government agencies, and non-profit organizations leverage to connect communities to education and information to advance their quality of life [23]. More recently, extension organizations have taken on a larger role in promoting health equity and community participation on a global scale, which is versatile and impactful.

Extension organizations, as an educational entity with a community focus, have significantly contributed to addressing health disparities and involving communities in the process. Recent guidance [22] has been instrumental in elevating health as a priority, promoting concepts like social determinants of health and a culture of health. This framework has led to collaborations between extension organizations and healthcare providers, resulting in research-based programs that empower communities to manage public health. Initiatives like “Well Connected Communities” have fostered local coalitions to address public health concerns, and the extension’s engagement has been recognized [22].

The extension organization’s commitment involves addressing the social determinants of health and adopting a targeted universalist approach, tailoring interventions to meet the needs of the communities with the greatest health burdens. Data-driven precision interventions, promoting healthy behaviors, and working with coalitions to create healthy communities are part of extension organizations’ new approaches, aiming to improve population health while achieving equity [22]. Notably, extension has transitioned from an expert-driven model to a participatory approach, where community members with lived experiences collaborate with professionals to address health disparities. This enables community members to become equal partners with agency professionals in the process of developing strategies and actions for community improvement [22]. This evolution in the approach to health equity in communities has led to significant changes within these communities which have previously faced notable disparities. Additionally, the involvement of community voices has led to many individuals in these areas feeling heard by the government and outside organizations.

The emphasis on addressing health inequities through extension organizations is growing [24,25,26,27,28,29,30,31,32,33,34,35,36] and highlighted in efforts such as that presented by Girmay et al., [37] in Addis Ababa, Ethiopia. In this example, the extension organization in Ethiopia prioritized public health in 2003 and facilitated the development and implementation of a comprehensive health transformation plan aimed at revolutionizing health service delivery. The extension organization facilitated a participatory approach to develop tailored components of the plan for urban and rural communities, in addition to the primary objectives of preventing diseases, promoting health, and implementing specific curative health measures [37]. From these efforts in Ethiopia, the public health outcomes have improved drastically over time due to the utilization of community-driven strategies such as creating accessible health centers and the development of health education programs focused on preventative health and first aid.

### 1.4. Purpose and Objectives

The purpose of this study was to identify a consensus on the most pervasive barriers and effective strategies to healthcare access and quality of care among refugees in Antioquia, Colombia. To achieve this purpose, the following objectives guided our inquiry:Determine the primary barriers to healthcare access and quality of care for refugees.Determine the most effective strategies to advance the healthcare access and quality of care for refugees.

This research study was developed to inform collaborative efforts between government agencies and officials in Antioquia, university partners, and community non-profit organizations that serve refugees and community leaders. The results of this study will be used to inform policies and practices that advance refugee health equity, including insights into the role of extension education.

## 2. Materials and Methods

In this study, we employed a modified, two-round Delphi technique, a process that involves anonymous questionnaires or surveys conducted iteratively with the aim of achieving consensus among a group of expert panelists in areas such as policy, practice, or organizational decision, within a relatively short time frame [38,39]. When applied in the context of community work, the Delphi technique facilitates open communication and collaborative decision-making, helping participants transcend individual biases and arrive at a collective perspective [40]. By anonymizing responses in the early rounds, it reduces the influence of dominant voices and allows for diverse viewpoints to be considered without fear of retribution or exclusion [39]. As the process unfolds, participants can modify their opinions based on the group’s feedback, which can lead to a refined and more informed consensus [41].

In community development and public health, the Delphi technique can serve as a powerful starting point for collective action. It helps participants identify common goals, priorities, and strategies, which are essential for effective community engagement and sustainable progress [22,23]. The method’s iterative nature allows for the exploration of complex issues from various angles, gradually revealing areas of agreement and disagreement. This iterative process enhances the shared ownership of decisions and actions, leading to a sense of empowerment and commitment among participants [42].

The panel of experts for this study comprised sixteen community health workers, specifically promotoras, who resided and worked in the economically deprived regions of the Medellin municipality, Antioquia department, Colombia, known as poor comunas or comunas populares, and who provided services to refugee communities (internally and externally displaced). These “promotoras” were chosen on purpose, due to their familiarity with the target communities and their healthcare needs, as well as their expertise in public health and healthcare. In addition, almost half of the population represented either internally displaced refugees or previous refugees from other countries.

Delphi studies usually have a predetermined number of rounds, and the key element is the iterative nature of the process, leading to consensus. For our study, we adopted a two-round approach, which is common in Delphi studies [43]. The research was conducted from the fall of 2022 to the spring of 2023, and all communication, both spoken and written, was carried out in Spanish.

We adapted the Delphi technique to include a minimum of two rounds, recognizing the limited availability of the panel members due to the challenges posed by their difficult home lives and extensive responsibilities in food production and community health. Beginning the collaborative effort from a point of consensus was crucial considering these constraints.

We focused on the populations of Antioquia due to a lack of research focused on this area, as Medellin has a big population of refugees that includes immigrants and displaced people due to internal conflict. The specific area of focus was the vereda or rural division of Granizal, which is a rural area outside of the municipality of Bello, next to Medellin. To choose the panelists, we adopted a community-based participatory research (CBPR) approach [44], seeking recommendations from key community leaders and non-profit organizations. As part of this approach, we formed an implementation workgroup comprising a subset of the panel (*n* = 5) to provide guidance throughout various aspects of the process, including a review and synthesis of the findings following the study.

### 2.1. Data Collection and Analysis

In the initial round of the two-round Delphi process, panelists were invited to participate in an online survey where they were asked to share their perspectives on four open-ended prompts. These prompts focused on identifying the most significant barriers to quality healthcare access for refugees, along with potential strategies for improving access to and quality of care. Before collecting data, the survey was piloted among a group of Colombian native speakers and extension educators in Florida to assess its readability and understanding. Using Colombians for the pilot ensured an evaluation based on the Colombian Spanish dialect and cultural norms. We asked four open-ended questions asked to panelists:Please list the most significant barriers, challenges, or obstacles to accessing healthcare for refugees.Please list the strategies that you believe hold the most promise for improving access to healthcare for refugees.Please list the most significant barriers, challenges, or obstacles to quality healthcare for refugees.Please list the strategies that you believe hold the most promise for improving quality healthcare for refugees.

To analyze the open-ended responses, a three-step thematic analysis process was employed [45]. Initially, two co-authors extracted relevant text related to the research questions from the respondents’ full answers to each prompt. Then, they grouped similar texts together within the areas of interest, namely, barriers and strategies. Next, the authors assigned a thematic label to each group of items that captured their common theme. This analysis was then shared with the rest of the author team and an external member to seek feedback. The process continued, incorporating the feedback, until all authors agreed upon the final themes.

Once these themes were set, a second round of questions was dispensed to the panelists so they could indicate the level of importance of each category, thus gauging the level of consensus among panelists. With regard to barriers, participants rated each obstacle on a four-point scale: (1 = not a barrier; 2 = somewhat a barrier; 3 = moderate barrier; 4 = major barrier), with consensus being defined as ⅔ of the panel designating it as a “major barrier”.

Regarding the strategies, participants were requested to evaluate the effectiveness of each proposed strategy using a five-point scale: (1 = not effective at all; 2 = somewhat effective; 3 = effective; 4 = very effective; 5 = extremely effective). Here too, we followed the established consensus standard, with 2/3 of the panel needing to designate a strategy as “Extremely effective” for it to meet the consensus threshold. Additionally, each list concluded with an open-ended query, inviting panelists to contribute any supplementary barriers or strategies worthy of consideration.

We decided that consensus was reached if at least 67% (⅔) of the panelists agreed with a barrier being a moderate barrier or major barrier, or with a strategy being moderately effective or extremely effective.

### 2.2. Study Limitations

Firstly, the Delphi technique, though designed to achieve expert consensus, can still be influenced by the selection of experts and their subjective perspectives. The study’s reliance on expert opinions may not necessarily represent the views of a broader population of immigrants. Secondly, the adaptation of a two-round Delphi process, while practical given the panel’s limited availability and challenging circumstances, may not provide the depth of consensus achievable in longer Delphi studies. Furthermore, the study’s focus on the Antioquia region of Colombia may limit the generalizability of its findings to other contexts. Additionally, using a community-based participatory research approach to select panelists might introduce biases.

## 3. Results

Table 1 shows that there were a total of eight potential barriers identified, of which seven were considered a “moderate or principal barrier” by our study’s standards of consensus; six of the barriers (barriers 1–6) reached a consensus of 80% or more, and one reached 77% (barrier 7). The barrier category that was not considered relevant by our panelists was that which identified xenophobia as a factor preventing the equality of healthcare services (barrier 8). On the other hand, the “lack of medical care centers that are refugee specific” was the barrier that received the most votes, identifying it as a principal one.

Table 2 shows that of the seven potential strategies identified by our panelists, all were identified as “very effective or extremely effective”, with five of them reaching a consensus of 80% or more (strategies 1–5), and two reaching a consensus of 77% (barriers 6 and 7). The strategy that was considered by most of our respondents as being “extremely effective” was that which proposed the creation of more local healthcare centers (barrier 6), while the strategy receiving the least number of votes in that section was the one that proposed the incorporation of a local cooperative system to aid in the navigation of the healthcare system (barrier 4).

Our panelists were presented with nine possible barriers to receiving quality healthcare (Table 3). Barriers 1–5 were identified as “moderate or principal barriers” by 80% or more of our respondents, and barriers 6–8 were considered to have the same identification by 77% or our respondents. Barrier 9 did not reach the minimum number of votes needed to be considered one of the main barriers, suggesting that the fear of discrimination is not a relevant limitation to quality healthcare, while barrier 1, lack of services for basic needs, received the most votes as a “principal barrier”, suggesting this is the most important one.

Our panelists were presented with nine possible barriers to receiving quality healthcare (Table 4). Barriers 1–5 were identified as “moderate or principal barriers” by 80% or more of our respondents, and barriers 6–8 were considered to have the same identification by 77% or our respondents. Barrier 9 did not reach the minimum number of votes to be considered as one of the main barriers, suggesting that the fear of discrimination is not a relevant limitation to quality healthcare, while barrier 1, the lack of services for basic needs, received the most votes as a “principal barrier”, suggesting this is the most important one.

There were a total of eight suggested strategies for tackling the barriers affecting the recipience of quality healthcare by refugees, and all of these were considered very effective or extremely effective with a consensus of 80% or more of our respondents identifying them as such. Barrier 3 (“Increase economic and technical support for healthcare promotion”) was identified as the most effective one by most of our panelists with 77% of them categorizing it as extremely effective.

## 4. Discussion

### 4.1. Barriers for Access to Healthcare

There were a total of eight potential barriers that were identified as affecting the access to healthcare, of which six were considered a “moderate or principal barrier” by at least 80% of our respondents. Many of the barriers presented are interconnected and it is evident that they work together to aggravate the lack of access to healthcare. For example, there is a lack of healthcare centers that are specific for refugee populations because the Colombian government has decided to focus on the potential inclusion of Venezuelan migrants into their already-present healthcare system instead of increasing the healthcare options available for these migrants outside of it [2]. Although this can be considered an innovative and inclusive approach, it limits care for immigrants that lack legal documentation.

Another barrier to accessing healthcare deals with the fact that “the most disadvantaged groups (those of low socioeconomic and educational levels) are the ones who are the least likely of being aware of the SGSSS” [11] (pg. 402), the General System of Social Security in Health. With most refugee populations having a low-socioeconomic status, they are at risk of not receiving the necessary information about the humanitarian healthcare resources available to them, as well as their rights to care. It was also identified by the panelists that the lack of internet access is a significant barrier to healthcare because that makes it harder to receive or research humanitarian help or any other medically related information.

The issue of unstable housing is one that affects Venezuelan migrants as well as internally displaced people who were forced to resettle due to problems in Medellin ranging from armed conflict to a lack of monetary resources. Many who have had to resettle do so in areas that are not legally approved for construction, due to factors such as how prone these places are to encountering natural hazards [4]. The possibility of experiencing a natural disaster that could destroy their home, as well as potentially being evicted due to the illegality of their residence is an issue that many refugees face. The house’s illegal status explains why the government does not help with the construction of better roads and transportation methods, which was an important barrier to healthcare access identified by the panelists.

### 4.2. Strategies for Access to Healthcare

As we analyze the strategies for access to healthcare, it becomes clear that all the proposed solutions are interconnected and usually the direct result of having identified a barrier. The creation of local healthcare centers, which relates to barrier 6 (Table 2), inevitably includes having a permanent or intermittent medical team in the community, and this inclusion of medical personnel would allow for medical workers to get a better grasp on the number of resources needed in the underserved communities of refugee populations.

The lack of medical resources, as an identified barrier, relates to one of the main strategies suggested, which was the addition of preventive health interventions, which in turn allows for less utilization of emergency care services. Those without legal documents are those most affected by the lack of preventative health, especially due to the conditionality of being seen in emergency care units; only those in extraordinary circumstances are treated, leaving those other major portions of the population untreated [8]. While the most immediate solution to this would be to help those populations acquire legal documentation, there is a clear issue with the lack of information about the ways in which to they can obtain the documents and then be able to navigate the healthcare insurance system—this reinforces the need for a support team to help refugees acquire legal documentation, followed by their acquisition of healthcare insurance, and a local cooperative system to help with navigation of the healthcare system.

A country that has experienced the benefit of the adoption of some of the aforementioned strategies is Costa Rica. During the last twenty years, Costa Rica has critically reformed its primary healthcare (PHC) system via the integration of multidisciplinary teams into communities, the increase in technology usage for keeping health records, and through geographic empanelment [46]. Letting communities take an active role in healthcare decisions allows for the implementation of better-fitting reforms, and this is what the communities of Costa Rica have experienced [47]. This community engagement framework to healthcare has brought many benefits to the health outcomes in this country, making it a potential model for others.

### 4.3. Barriers to Receiving Quality Healthcare

There were a total of nine potential barriers to quality healthcare identified, of which eight were considered a “moderate or principal barrier” by at least 80% of our respondents. Many of the barriers presented are interconnected and it is evident that they work in conjunction to affect the quality of healthcare within the community.

The main barrier identified by community leaders was the lack of services for basic needs such as potable water, sanitation services and access to food stores. The lack of clean drinkable water is a particularly serious concern, as access to clean water is essential for health and hygiene. According to the International Water Association, as of 2020, there is a 85% coverage for urban drinking water and 81% coverage of urban sewerage in these communities [48]. However, it is important to note that these percentages mask large gaps and inequalities in these areas. The lack of potable water is a major issue that the Colombian government and non-profit organizations have recognized and have made attempts to solve by creating programs that bring clean potable water to these underdeveloped regions via water trucks—these water trucks would ideally bring clean water to these areas and store the water in retention tanks for shared usage in the whole community [49]. Although this is a solution to the issue at hand, several other issues have arisen such as the lack of infrastructure required for these water trucks to reach these areas, as well as the contamination of the retention tanks by mosquitoes, larvae, and other insects [50]. Furthermore, the lack of infrastructure regarding the building of roads and sanitation services in these regions is a significant barrier to the quality of health and overall wellbeing. The lack of infrastructure has been documented in these regions as due to them being unofficial settlements formed out of necessity for refugee communities [4,51]. The deficit of established roadways in these regions significantly impacts the ability of individuals to travel to reach the established parts of the city where they can receive healthcare and purchase food for their families. The fact that many people do not have access to clean water and proper sanitation facilities underscores the need for urgent action to improve living conditions and public health. Improving access to clean water and sanitation can have a positive ripple effect on various aspects of life, including health, education, and overall wellbeing.

Additionally, the inability to acquire a healthcare insurance plan, insufficient coverage by insurance plans, and inconsistent medical care due to a lack of legal status were all identified as principal barriers. These barriers are not only seen as barriers to access but are consistently identified in the literature as a pervasive barrier to refugee healthcare. The absence of proper legal documentation gives rise to a host of issues concerning the inconsistent medical care received by refugee populations, primarily due to their unsettled migratory status. This lack of documentation is rooted in a broader lack of understanding about the procedures for obtaining legal status. Without acquiring this legal status, migrants find themselves locked out of the possibility of enrolling in healthcare insurance plans.

Consequently, the inadequacies in the coverage provided by insurance plans can be traced back to this divide between the legal and actual extents of their coverage. This divide becomes starkly evident when comparing the accessibility of basic healthcare between individuals residing in prosperous regions and those inhabiting lower-income areas, as highlighted by Lamprea and Garcia [52].

As a result of no or insufficient coverage, Colombia’s private and public hospitals often refuse to provide care to individuals, citing the overwhelming surge in patient numbers as a justification. If not an immediate emergency, groups such as older adults, men, non-pregnant women, individuals dealing with chronic ailments, those with non-life-threatening conditions, and those in need of mental health assistance, find themselves excluded from the coverage provided by this emergency care framework [8]. Consequently, this scenario frequently leaves numerous migrants with unaddressed chronic health issues. Most Venezuelan migrants find themselves devoid of legal documentation, rendering them uninsured and requiring them to personally cover the costs of essential health services.

Lastly, self-diagnosis and self-medication practices were identified as a principle or moderate barrier by 67% of the participants. In these communities, it is very common to see individuals choosing to find alternative medicine solutions to their illnesses because of the barriers preventing them from seeking quality healthcare [53]. Alternative medicine options such as self-medicating in pharmacies, telemedicine from doctors abroad, reliance on uncertified Venezuelan doctors, and the use of social media to exchange knowledge about care, have been documented in these populations [8]. Self-medication practices including purchasing medications at pharmacies and utilizing common herbs and household items in place of medication are extremely common. This can often cause more harm than good for individuals and can lead to dangerous health emergencies without the guidance of a healthcare professional diagnosing and prescribing the correct treatment plan. Furthermore, many migrants prefer non-biomedical health services, which highlights the importance of trust and cultural preferences with regard to healthcare [8].

It is interesting to note that the fear of discrimination was not identified as a barrier to healthcare quality. It has been documented previously that those who have experienced forced displacement are likely face discrimination, stigmatization, racism, socio-cultural differences, and economic marginalization [4,8]. Additionally, there are instances of Venezuelan migrants being completely turned away from services due to xenophobia, playing a direct role in not only quality but the overall access to healthcare as well [44]. Although discrimination can take various forms, in our study, the participants have not faced discrimination regarding accessing quality healthcare.

### 4.4. Strategies for Receiving Quality Healthcare

There were eight strategies to improve the quality of healthcare that were deemed extremely and very effective by consensus from our study sample (see Table 4). A strategy deemed effective for the refugee population was to create a mobile healthcare team for rural areas, as they are seen as a strategy to create access to and quality of care. According to Garcia, geographical barriers affect those who are under a contributory insurance scheme, specifically those that are lower income [54]. Geographical barriers make it much more difficult for people to reach healthcare centers, no matter whether they are insured or uninsured. Using a mobile healthcare team flips the paradigm by bringing healthcare to the communities that most need it.

As mentioned in the section on access to healthcare, the public health system in Costa Rica has created a system that brings primary healthcare directly to its patients. This system integrates rural health centers and multidisciplinary teams that visit patients based on their level of need, while also being cost-effective. This plays a major role in decreasing health disparities, because not only is it an increase in access, but with more primary care visits there is less of a reliance on emergency services [46]. There is potential for a version of this strategy to be applied to the Colombian healthcare system, hopefully yielding similar results.

The second most impactful strategy involves the reduction of wait times between general practitioners and medical specialists. As discussed earlier, several barriers elucidated in our study, including issues like inconsistent medical care and inadequate medical coverage, have the potential to amplify these waiting periods. The lack of consistency makes it challenging for patients to sustain a proper diagnosis, maintain ongoing monitoring of their progress, and secure access to necessary medications [55].

The array of agreed-upon strategies for enhancing healthcare quality encompass several key approaches. These strategies include bolstering economic and technical support for healthcare promotion, intensifying healthcare education pertaining to proactive health measures, and providing comprehensive insights into the broader healthcare system. All of these strategies collectively fall under the overarching aim of educating the local population about the available resources, thereby facilitating smoother navigation of the system.

In line with this notion, Eigner’s research on Venezuelan refugees in Colombia underscores a critical point: a significant lack of awareness exists regarding the country’s healthcare system—its prerequisites, protocols, as well as the entitlements and support channels accessible for medical care [56]. This observation directly aligns with our study’s findings, reinforcing the assertion that a generalized information deficit is pervasive among the refugee population. The absence of proper education on available resources and their effective utilization diminishes the impact of these resources considerably.

A fundamental requirement emerges for an enhanced health education initiative targeted at the refugee population. Our study’s respondents concur that these proposed strategies hold the most promise in addressing the situation, thereby charting a clear direction for future research endeavors. This information applies not only to refugees but also extends to internally displaced communities and Venezuelan immigrants, who share the same challenges. The common thread is a conspicuous dearth of information reaching these groups, rendering access to healthcare an arduous task.

Through this research, a tangible understanding emerges of the strategies that the refugee population identifies as potentially transformative. These strategies offer a starting point for unraveling the complexities of this issue.

## 5. Conclusions

By drawing upon the collective consensus of sixteen community leaders within the greater Medellin and Bello region, distinct and recurrent barriers that wield significant influence over the overall health of rural populations have come to light. The utilization of the Delphi technique has not only enabled educators to coordinate their collaborative efforts strategically but has also furnished them with a thoughtfully prioritized compilation of barriers and potential strategies to contemplate.

Several notable barriers impeding healthcare access have been unearthed, a consensus among the respondents is that of the absence of fundamental transportation infrastructure and localized healthcare facilities and resources. These deficits have emerged as pivotal contributors to the deficiency of adequate care for these underserved populations. Beyond these, barriers surrounding the quality of healthcare have also gained prominence. Elements such as inconsistent medical care, a lack of familiarity with humanitarian assistance, and challenges surrounding health insurance accessibility, have all been pinpointed as critical determinants affecting the refugee population residing in these rural settings.

The outcomes gleaned from the Delphi study have paved the way for a thorough exploration of the various strategies aimed at overcoming these barriers. Given the inherent challenges in delivering resources to rural locations, strategies focused on bolstering awareness regarding legal status attainment, the acquisition of health insurance, and the propagation of preventative care education, have emerged as pragmatic interim solutions for aiding the rural refugee community. The implementation of enduring primary healthcare reform strategies, similar to the health policies discussed in Costa Rica, would provide lasting change in rural healthcare in the long term. While more enduring strategies like the establishment of secure infrastructure and local healthcare centers hold promise, their implementation requires a lengthier planning and funding trajectory.

Furthermore, there is an evident deficiency in education regarding the acquisition of essential resources and the navigation of the intricate bureaucratic landscape of healthcare. This knowledge gap exacerbates the challenges these populations face in accessing high-quality medical care. Addressing this informational void concerning legal document acquisition and healthcare navigation holds the potential to substantially alleviate health disparities within these rural communities.

The interconnectedness of the data gleaned from our study reveals a compelling narrative: that the amelioration of pivotal issues can trigger a ripple effect, thereby positively influencing a multitude of other challenges. Through cooperation with extension educators and collaborations extending beyond the academic sphere, abundant opportunities exist to address these longstanding health disparities through education and holistic community development.

## Figures and Tables

**Table 1 ijerph-20-06948-t001:** Summary of panelists’ ratings for each barrier to healthcare access as a moderate or principal barrier by number and percentage.

Barrier	Principal Barrier (%/n)	Moderate Barrier (%/n)
Lack of medical care centers that are refugee specific **	69.23 (9)	30.77 (4)
Lack of internet connection to access medical information **	61.54 (8)	30.77 (4)
Lack of monetary resources *	53.85 (7)	23.08 (3)
Bad transportation **	53.85 (7)	46.15 (6)
Lack of knowledge about available humanitarian help **	53.85 (7)	38.46 (5)
Lack of legal documentation **	38.46 (5)	53.85 (7)
Unstable housing **	30.77 (4)	69.23 (9)
Xenophobia preventing equality of healthcare services	23.08 (3)	38.46 (5)

** >80% consensus. * >67% consensus.

**Table 2 ijerph-20-06948-t002:** Summary of panelists’ ratings for each strategy to aid healthcare access as extremely and very effective by number and percentage.

Strategy	Very Effective	Extremely Effective
Support to get legal documentation **	76.92 (10)	23.08 (3)
Preemptive health interventions **	46.15 (6)	46.15 (6)
Have a permanent medical team in the community **	61.54 (8)	30.77 (4)
Local cooperative system for navigation of the healthcare system **	84.62 (11)	7.69 (1)
Acquisition of healthcare insurance **	61.54 (8)	23.08 (3)
Create local healthcare centers *	23.08 (3)	53.85 (7)
Preemptive health education *	30.77 (4)	46.15 (6)

** >80% consensus. * >67% consensus.

**Table 3 ijerph-20-06948-t003:** Summary of panelists’ ratings for each barrier to quality healthcare as a moderate and principal barrier by number and percentage.

Barrier	Moderate Barrier	Principal Barrier
Lack of services for basic needs (e.g., potable water) **	7.69 (1)	76.92 (10)
Inconsistent medical care due to legal status (mobility problems to healthcare centers) **	46.15 (6)	53.85 (7)
Lack of knowledge about humanitarian help available **	38.46 (5)	53.85 (7)
Insufficient coverage by insurance plans leaves many illnesses and medical emergencies untreated for **	30.77 (4)	53.85 (7)
Lack of acquirement of an accessible healthcare insurance plan **	53.85 (7)	38.46 (5)
Lack of medical insurance *	30.77 (4)	46.15 (6)
Lack of legal documentation *	30.77 (4)	46.15 (6)
Self-diagnosis and self-medication practices *	30.77 (4)	46.15 (6)
Fear of discrimination	15.38 (2)	7.69 (1)

** >80% consensus. * >67% consensus.

**Table 4 ijerph-20-06948-t004:** Summary of panelists’ ratings for each strategy to advance quality healthcare access as extremely as very and extremely effective by number and percentage.

Strategy	Very Effective	Extremely Effective
Create a mobile healthcare team for rural areas **	30.77 (4)	69.23 (9)
Reduce the wait time for general and specialized visits **	30.77 (4)	69.23 (9)
Increase economic and technical support for healthcare promotion **	15.38 (2)	76.92 (10)
Increase health education about preemptive health **	30.77 (4)	61.54 (8)
Humanization of patients by the healthcare providers **	46.15 (6)	46.15 (6)
More access to healthcare system information **	84.62 (11)	7.69 (1)
Provide legal advice on healthcare service paperwork **	53.85 (7)	30.77 (4)
Technical audits to evaluate the quality of healthcare services **	53.85 (7)	30.77 (4)

** >80% consensus.

## Data Availability

The data presented in this study are available on request from the corresponding author. The data are not publicly available due to privacy considerations.

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
