# Peer review of "Evaluating Rural Health Disparities in Colombia: Identifying Barriers and Strategies to Advancing Refugee Health"

_ijerph, 2023, doi:10.3390/ijerph20206948_

Round 1

Reviewer 1 Report

This study used a two-round Delphi technique to gather data about the barriers and strategies to healthcare among Venezuelan refugees in Colombia .  This study is important and the abstract reads well; however, I have some recommendations for this paper that could improve it further:

1.     Please also consider the international organization for migration for up to date statistics for the introduction section. 

2.     Please elaborate on the meaning of irregular migrants.

3.     Perhaps state that Venezuela is a neighboring country in the introduction section (although you mention it later, it might be better to shift it here).  

4.     Line 40 needs to specify which country’s population – Venezuela or Colombia?

5.     Section 1.1 – perhaps mention housing as a social determinant of health?

6.     Line 84 – What is the nature of the physical barriers? Who set up the barriers?

7.     No data on the validity of the questions or questionnaire used to gather the physical activity data. 

8.     Methods – survey methods using two-round Delphi technique – iterative process that led to a consensus (defined as 2/3 of the participants/panelists identifying the issue to meet threshold requirement) – good.

9.     Methods – sampling was indicated as purposeful – good.

10.  Methods – sample size was 16, this is considered low.  

11.  Analysis – thematic analysis – good

12.  Line 315 suggests that the barriers are numbered or ranked but this is not clear from Table 1.

13.  What are the decimals and parentheses represent in Table 1? Same for Tables 2 to 4. Each table should have an explanation of these values e.g. Likert scale rankings?

14.  Discussion 4.1 -  low socioeconomic status and low educational attainment was not identified as a barrier in Table 1 and 3 so I wonder how this statement suggests that?

15.  Line 400 states medical resources but I think it would be appropriate to state monetary resources for accuracy?

16. Strengths and limitations – Good to see the purpose of the research; What are the limitations?

17. Other issues (minor) – referencing format does not conform to the journal’s requirements for numbered endnotes style

18. Other issues (minor) – 2/3 is actually rounded up to 67%, not 66% - consider revising this on lines 309 and 322

19. Table 1 – perhaps organize the barriers by the ranking

 Thank you for the opportunity to review this manuscript.

Reviewer 2 Report

This study looks at the barriers to healthcare access for migrant communities in Columbia.  The authors appropriately emphasized the importance of this topic as the world continues to grow more interconnected and the number of people migrating continues to grow.  The authors also did a fine job situating their research in the academic discourse and utilizing the literature review to generate their research questions and gaps in the academic literature. 

My primary concern surrounds the sampling technique the authors used for their questionnaire.  The focus on health care leaders provides one half of the story of the challenges facing migrant communities.  The authors do not include the migrants themselves as part of the sample.  This would provide a more complete picture of the healthcare situation in Columbia. 

While I think the lack of migrants voices in your sample does limit the conclusions you can generate, I don't see this as a reason to reject the research outright.  If this was my research, I would qualify my conclusions to address the lack of migrant voices in your sample population. 

Overall, I think this is a critical and timely topic that deserves the attention you are providing.  I think you did a fine job situating your research into the larger academic discourse.  This will generate greater interest for your research beyond the healthcare community into researchers and policy makers looking at migration and refugeeism.  Thank you for the opportunity to review your research.  I learned a lot from reading your article.  Best of luck in your future research.    

I think a proofread of the final version would be helpful.  I found a few run on sentences that effected readability a bit. Overall, it was well written.  
